# GENETIC ALGORITHM FOR CURRICULUM DESIGN IN MULTI-AGENT REINFORCEMENT LEARNING

## ABSTRACT

As the deployment of autonomous agents increases in real life, there is an increased interest in extending their usage to competitive environments populated by other robots. Self-play in Reinforcement Learning (RL) allows agents to explore and learn competitive strategies. However, the complex dynamics of multi-agent RL interactions introduce instability in training and susceptibility to overfitting. Several game-theoretic approaches address the latter by generating approximate Nash equilibrium strategies to train against. The challenge of learning a policy in a complex and unstable multi-agent environment, the former, is not yet well addressed. This paper aims to address this issue by using a curriculum learning approach. We introduce curriculum design by a genetic algorithm to the multi-agent domain to more efficiently learn a policy that performs well and is stable at Nash equilibrium. Empirical studies show that our approach outperforms several strong baselines across various competitive two-player benchmarks in continuous control settings.

## 1 INTRODUCTION

Competitive multi-agent reinforcement learning has attracted attention for its potential in real-world applications such as gaming, robotics, sports, finance, and cybersecurity, where agents compete against each other. In Reinforcement Learning (RL), self-play helps agents become competitive by exploring and discovering new strategies on their own. Although this kind of training should ideally produce an agent capable of performing well across various scenarios and against different opponents, self-play agents often achieve suboptimal outcomes. The complex dynamics of multi-agent training make it easy for the agent to settle into a local minimum. Agents tend to exploit specific weaknesses in the opponent's policy observed during training rather than striving for the global optimum. This leads to a slow rate of convergence throughout the training and a migrating policy problem in the later phases of the training.

One way to mitigate the issue of a migrating policy is to train against a population of opponent policies to avoid overfitting to a single opponent policy (Heinrich et al., 2015; Heinrich & Silver, 2016; Vinyals et al., 2019). Various game theoretic approaches (Lanctot et al., 2017; Smith et al., 2020; Feng et al., 2021) use approximate Nash strategy opponents to maintain theoretical stability near the equilibrium. A Nash equilibrium (Nash Jr, 1950) is the point where none of the players can improve their strategies (represented as a mixture of policies) to improve their outcome. Training against an approximate Nash strategy ensures stability once reached. However, such works often do not address the challenge of navigating towards the Nash equilibrium in a highly unstable training environment caused by complex multi-agent interactions.

In a single-agent RL domain, one way to stabilize and enhance the rate of convergence is to use curriculum learning (Asada et al., 1996; Karpathy & Van De Panne, 2012; Held et al., 2018; Florensa et al., 2017; 2018; Narvekar & Stone, 2019a; Fournier et al.; Matiisen et al., 2019). By gradually introducing similar tasks around the boundaries of the agent's performance with the highest learning potential, curriculum learning has shown faster convergence to a better solution.

Inspired by these findings, we focus on utilizing curriculum learning for a game-theoretic setup. During training, our curriculum generator will generate scenarios and opponents just outside the boundary of the agent's expertise. Over time, those opponents will evolve toward the Nash equilibrium. We identify three algorithmic innovations that lead to our improved performance: The use of 1)

population-wide genetic operations (crossover) rather than simple modifications of the replay buffer (random sampling / mutation); 2) regret to tailor the difficulty level of genetically generated scenario, and 3) continuously optimized open-loop opponents to stabilize early learning. We include an ablation study to highlight the effects of our assumptions and design features. Figure 1 shows the overall structure of our algorithm.

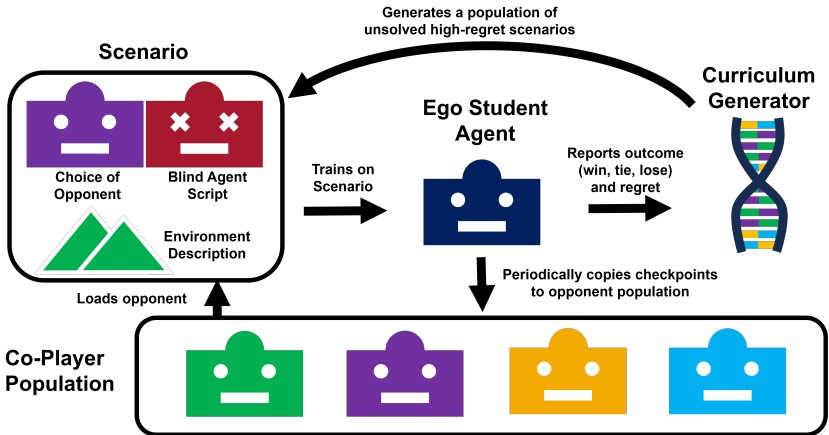

Figure 1: Overview of our proposed approach: During training, the ego student agent is trained against a scenario describing the choice of opponent, action script for the blind agent, and the choice of environment parameters. Our curriculum keeps a record of regret and win/lose/tie outcomes for each scenario. Based on the performance of the scenarios in the previous population, our curriculum generator uses a genetic algorithm to generate a new population of scenarios to be used to train the agent for an epoch at each fixed interval of steps.

## 2 RELATED WORKS

**Game Theoretic Approach to Self-Play**

Training an RL agent in a competitive environment requires an opponent to train against. Unlike using expert demonstrations (Chen et al., 2021) or rule-based opponents (Burgin & Sidor, 1988), self-play (Tesauro et al., 1995; Silver et al., 2018) provides a unique opportunity. As the ego agent improves and explores various parts of the game, the opponents' proficiency and understanding of the game also improve, enabling agents to discover new competitive behaviors autonomously.

However, some drawbacks of self-play includes 1) the instability of training against an opponent that is changing over time (Garnelo et al., 2021) and 2) a tendency to overfit to a spefic opponent rather than learning a policy that generalizes across various opponents and situations. While some approaches address this challenges with expert supervision (Won et al., 2021), this can be costly.

The approach commonly employed in self-play, as proposed in Fictitious Self-Play (FSP) (Brown, 1951; Leslie & Collins, 2006; Heinrich et al., 2015), involves training against a population of policies by saving thecheckpoints of the ego policy during training. Expanding on these findings, game theoretic approaches like Policy Space Response Oracle (PSRO) (Lanctot et al., 2017; Vinyals et al., 2019; Berner et al., 2019) utilize Nash equilibrium (Nash Jr, 1950), a stable point in a multi-agent game where no player can update itself to improve the outcome. PSRO calculates an approximate Nash strategy to determine the mixture of checkpoints to load as opponents. However, finding an approximate Nash strategy can be computationally costly. Moreover, the difficulty of learning a policy with complex multi-agent dynamics is still not well addressed. While some approaches explore ensemble learning to split the learning task (Smith et al., 2021) (Smith et al., 2020), they are mostly limited to simple problems with discrete action spaces.

**Curricular Reinforcement Learning**

In the single-agent domain, addressing the challenge of learning a difficult task is often tackled through Curricular RL. Curricular RL suggests that rather than training the agent directly on a

challenging task, it should first be exposed to simpler tasks or scenarios and progressively introduced to slightly more complex but similar ones to learn faster. (Florensa et al., 2018; 2017; Narvekar & Stone, 2019b; Ivanovic et al., 2019; Klink et al., 2020; Portelas et al., 2020; Dennis et al., 2020).

Curriculum optimization can be done in various ways, such as Bayesian Optimization (Paul et al., 2019) or teacher agents (Du et al., 2022). Some works (Wang et al., 2019; 2020) explored how to use genetic operations for curricular RL. By generating scenarios similar to the scenarios that taught the agent well, genetic operations can continuously generate candidates of scenarios that will teach an agent well.

While most curricular RL approaches using genetic algorithms rely on mutations, which make small alterations to a scenario's encoding, Genetic Curriculum (GC) (Song & Schneider, 2022) investigated the use of population-wide genetic operations, such as crossover, in curriculum generation. Crossover involves merging encoding sequences across a population of scenarios, facilitating the transfer of skills within the curriculum by enhancing simlarity among the scenarios. However, GC is computationally intensive due to the evaluation steps necessary for curriculum generation. Additionally, GC lacks a true means to regulate difficulty.

Regret is often used in curricular RL to regulate difficulty. Quantifying the gap between optimal and actual performance of an agent, regret provides a valuable measure of the agent's improvement potential. Showing success in various single-agent domains Jiang et al. (2021a); Parker-Holder et al. (2022), regret has been expanded to do curriculum learning in multi-agent setup by Multi-Agent Environment Design Strategist for Open-Ended Learning (MAESTRO) (Samvelyan et al., 2023). While MAESTRO also introduced optimizing both environmental parameters and choice of opponents to guarantee robustness, it is limited due to using domain randomization to discover new scenarios. Studies in GC Song & Schneider (2022) suggests that this can lead to suboptimum performance. As randomly generated scenarios will not be similar to each other as ones generated via population-wide genetic operations such as crossover, transferring skills between scenarios will be difficult, leading to a slower learning process.

**Open-Loop Opponents for Multi-Agent Self-Play**

Self-play often grapples with instability, particularly at the outset of training when the opponent is not sufficiently trained to make meaningful actions in the game. While some approaches incorporate hand-crafted opponents (Vinyals et al., 2019) or agents trained via imitation learning from expert data (Won et al., 2021), such supervision can be expensive to prepare.

One approach to providing remove expert supervision for an opponent agent is by utilizing a No-OP agent (Team et al., 2023), which takes no actions. However, these agents cannot generate complex responses and are impractical in environments where action is essential to remain in the game. For instance, a walking robot will fall if there's no torque in the joints. Plane with no control input will eventually crash by losing speed and altitude due to drag.

## 3 APPROACH

This section covers our approach for using population-wide genetic operations with regret-based difficulty regulation and continuously optimized open-loop opponents to help agent perform well in a competitive multi-agent environment.

### 3.1 PRELIMINARIES

A RL problem setup is typically represented as a tuple in a Markov decision process: $[S, A, P, r, \gamma]$, where $S$ is the state space of a problem, $A$ is the action space, $P$ is the transition dynamics, $r$ is the return of a state-action, and $\gamma \in [0, 1)$ is the temporal discount factor. The agent's policy, $\pi(a \mid s)$, maps states $s \in S$ to actions $a \in A$. The utility of a policy $\pi$ is the expected return, $J(\pi) = \mathbb{E}a_t \sim \pi \sum t \gamma^t r(s_t, a_t)$. During training, an RL algorithm optimizes the policy with respect to the data it collected about the reward and state dynamics.

In our multi-agent setup, we consider that the utility of a policy is also dependent on the opponent's policy $\pi_{opp}$ and the environment $\psi$, denoted as $J(\pi, \pi_{opp}, \psi)$. At each epoch consisting of a fixed number of time steps, we save the current version of our ego agent and add it to the library of

possible opponents to choose from, indexed by the integer $\pi_{opp}$. We descript the environment with environmental parameters, $\psi$. We define a scenario $\xi$ as an opponent policy - environment pair: $\xi = \{\pi_{opp}, \psi\}$.

To avoid overfitting to a single policy, we train our agent against a population of opponent policies. Therefore, we design our curriculum generator to optimize both $\pi_{opp}$ and $\psi$ when generating a curriculum.

Following this approach, we attempt to generate a curriculum consisting of a population of a fixed number of scenarios, $\Xi_{train} = \xi_0, \xi_1, ..., \xi_n$, where optimizing the ego policy $\pi$ with respect to the curriculum $\Xi_{train}$ will result in a policy with the behaviors we desire.

$$\pi^* = \max_{\pi} J(\pi, \Xi_{train}) \tag{1}$$

## 3.2 Problem Formulation

Following a game-theoretic approach, we set our curriculum generator to define the mixture of opponents to play against. To run our optimizer, we need to define what we would like to minimize.

We reformulate the game as a zero-sum game $G(\pi, \xi)$ where $G(\pi, \xi) = 1$ if $\pi$ wins, $G(\pi, \xi) = 0$ if it was a tie, and $G(\pi, \xi) = -1$ if loses. The opponent $\pi_{opp}$ and the environment parameters $\psi$ are determined by the curriculum's scenario $\xi$. The solution for a finite zero-sum 2 player game is minmax;

$$G^* = \min_{\Xi} \max_{\pi} G(\pi, \Xi) \tag{2}$$

Therefore, a population-based curriculum generator should generate a population $\Xi$ that minimizes $G(\pi, \Xi)$

## 3.3 Genetic Algorithm for Curriculum Generation

We utilize a genetic algorithm as our curriculum generator. Genetic algorithms are well-suited for generating curricula due to their ability to produce scenarios that inherently similar each other, and their flexibility in scenario encoding during training.

At the beginning of the training, we randomly initialize $\Xi$. $\pi$ is trained on the $\Xi$ for an epoch and records whether it has won or lost or tied, along with the approximate regret estimated by positive value loss (Jiang et al., 2021a;b).

We then harvest $\xi$s and use crossover and mutation (see Figure 2) to create an offspring population consisting of sequences of $\xi$ similar to the ones harvested. Crossover occurs when we take a random segment from one parent scenario encoding and swap it with a random segment from another parent scenario encoding. Mutation is when we select a random segment from a parent scenario encod-

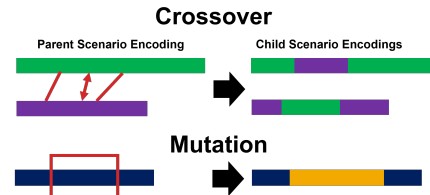

**Figure 2:** Visualization Crossover and Mutation Operations. Crossover is performed by replacing encoded segments between two parent scenarios. Mutation is performed by changing a encoded segment of a parent with a random sequence.

ing and swap it with a segment from a randomly generated scenario encoding. Since our aim is to minimize $G(\pi, \Xi)$, the fitness function (representing the likelihood of a $\xi$ being harvested, denoted as $p(\xi)$ for parents for crossover) is set as $p(\xi) \propto (1 - G(\pi, \xi))$. Detailed operations for crossover and mutation can be found in Appendix A.4.1.

Scenarios $\xi$s generated by the genetic algorithm will inherently be similar to each other, aiding in the transfer of skills and consequently a faster rate of convergence when used as a curriculum. However, a genetic algorithm alone cannot directly regulate the difficulty level of the generated scenarios. This is where regret comes into play. As regret measures the information potential of a scenario [?], aiming for high-regret scenarios allows us to regulate the difficulty level by sampling scenarios where $\pi$ has the biggest room for improvement. Therefore, we set $p(\xi) \propto \delta(\xi)(1 - G(\pi, \xi))$.

### 3.4 Blind Agent and Scenario Space

Most approaches in multi-agent self-play require means to avoid overfitting to local minima as RL opponents are often susceptible to adversarial attacks. Instead of searching for the global optimum, the student agent will often exploit the adversarial weakness of the opponent by visiting states where the specific opponent policy is noisy and has failed to generalize and perform well. We compensate for this effect by introducing a Blind Agent, denoted as $\pi_\varnothing$, which cannot be exploited in this manner.

Instead of encoding the opponent as an index on which checkpoint to load, $\pi_\varnothing = \{(t_0, a_0), (t_1, a_1), (t_2, a_2).....\}$ encodes the opponent as a list of non-fixed lengths describing which actions to take in an open-loop fashion. For example, at timestep $t_1 < t < t_2$, the opponent takes action $a_1$.

Blind Agents, running in an open loop without state observation, cannot be exploited like RL opponents. This prevents ego agent overfitting by avoiding confusing actions to exploit a specific opponent policy. Instead, it encourages learning a general policy resilient to opponent weaknesses. As genetic algorithms can optimize sequences of non-fixed length, $\pi_\varnothing$ is generated and evolved during training without expert supervision. Incorporating all these features, we encode a scenario as $\xi = \{i_\pi, \pi_\varnothing, \psi\}$

$i_\pi$ is an integer representing the i-th checkpoint to load as the opponent $\pi_{opp}$. If $i_\pi = 0$, it indicates that no agent will be loaded, and instead, a Blind Agent $\pi_\varnothing$ will be used as the opponent. $\psi$ denotes which environment parameters to load. It's important to note that whether it is used or not, the encoding for the Blind Agent $\pi_\varnothing$ is always present in a scenario $\xi$. This ensures that the optimization and the memory about a Blind Agent $\pi_\varnothing$ are not lost during the genetic operations and can always be brought back if needed throughout the training Algorithm 1 shows the pseudocode of our proposed algorithm, GEMS.

---

**Algorithm 1** GEnetic Multi-agent Self-play (GEMS)

---
1: **Initialize** Policy $\pi$
2: #*Select Curriculum Size and Mutation Rate*
3: **Input** $M_{train}, p_\mu$
4: #*Initialize Curriculum of size $M_{train}$*
5: **Initialize** Curriculum $\Xi_{train}$
6: **while** True **do** outcome = [] regrets = []
7:     #*Train $\pi$ with $\Xi_{train}$ by exploring scenario $\xi$*
8:     **for** $\xi$ **in** $\Xi_{train}$ **do**
9:         $u, \delta = $ Train$(\pi, \xi)$
10:         outcome.append($u$)
11:         regrets.append($\delta$)
12:     **end for**
13:     #*Harvest Examples*
14:     $\Xi_{seed}$, utility = **harvest**($\Xi_{train}$,outcome,rewards)
15:     #*Generate New Curriculum*
16:     $\Xi_{train} = $ **crossover**($\Xi_{seed}, M_{train}$,utility)
17:     $\Xi_{train} = $ **mutation**($\Xi_{train}, p_\mu$)
18: **end while**

---

## 4 Experiments

### 4.1 Benchmarks

We include three benchmarks with varying complexity and game dynamics to showcase our algorithm's performance against diverse baselines across different problems. Further environment details can be found in Appendix A.1

***Pong*** is a 2-player, continuous-action space version similar to Atari Pong (Brockman et al., 2016). In this game, each player controls a paddle on the left and right, attempting to hit the ball towards the opponent's side to win. The scenario describes which player controls which side and the initial velocity of the ball. This is the simplest of all environments, as each paddle can only move in 1-D space (up and down) and the ball's horizontal speed is fixed, simplifying the physics and making it relatively easy to learn. Therefore, this environment primarily emphasizes the game-theoretic aspect of the approaches compared to other benchmarks.

***Volley*** is based on (Hardmaru, 2020). Each player controls an avatar controlled continuous action input to move in left, right, and jumping. The objective is to bounce the ball to land on the other side of the map across the net. The scenario specifies the player's side and the initial ball velocity. This environment is more complex to learn than Pong due to the the presence of gravitational pull and elastic collision physics, elements absent in Pong. This complexity demands a stronger focus on acquiring proficient skills through curricular learning.

***ACM*** is a simulated dogfighting environment. In this setting, each player commands an airplane in 3D space, aiming to position its nose toward the opponent without crashing to the ground. The scenario describes the positions, postures, and velocities of the spawning airplanes. Among the three benchmarks, this environment is the most complex. The aircraft operate in 3D space with direct control over aerodynamic surfaces, devoid of stability assistance. The physics are simulated using JSBSim (Berndt, 2004), a high-fidelity simulator widely adopted in autonomous aircraft and aircraft controls research due to its accurate aerodynamic modeling (Pope et al., 2021). Consequently, a successful baseline should guide agents in solving challenges within this intricate environment. Additionally, since the game's objective is to maneuver and position the ego plane in relation to the opponent, the scenario—encoding spawn locations and orientations of the planes—significantly influences the game's outcome.

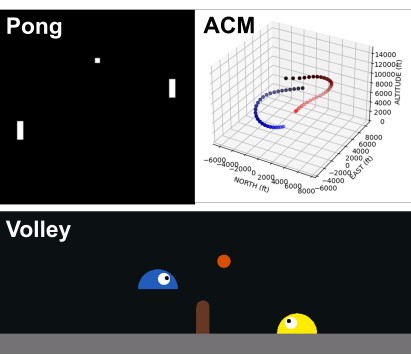

Figure 3: Screenshot of Benchmarks used in this paper, Pong, Volley, and ACM (aircraft trajectories are marked in blue and red)

## 4.2 BASELINE ALGORITHMS

We choose PSRO as one of the baselines for comparison. While numerous algorithms have emerged from PSRO, including parallel approaches like Rectified PSRO (Balduzzi et al., 2019) and Pipeline PSRO (McAleer et al., 2020), or those aiming to find meta-solutions minimizing exploitability such as Anytime PSRO (McAleer et al., 2022), these approaches are fundamentally based on PSRO's framework. They all share the underlying structure of generating Nash strategy opponents without considering ego agent's performance to present the tasks in a gradual, easy to learn fashion. Therefore, we utilize PSRO as a characteristic example to highlight the limitation of this assumption. While PSRO necessitates additional steps to evaluate each policy in order to run the meta-solver during training, we report performance without considering the extra evaluation steps that PSRO requires. We believe that presenting PSRO's performance without factoring in these additional steps provides a comprehensive comparison, not only against PSRO itself but also against newer subsequent algorithms such as XDO (McAleer et al., 2021) and NAC (Feng et al., 2021), which aim to reduce the computational cost associated with policy evaluation.

We include GC (Song & Schneider, 2022) as a comparison against approaches that uses a genetic algorithm to generate a curriculum. We include SPDL (Klink et al., 2020) to compare our approach against algorithms controlling environment parameters and actively regulating difficulty levels during curriculum generation. For fairness in multi-agent domain, GC and SPDL are running with Fictitious Self-Play (FSP) (Heinrich et al., 2015),labeled GC+FSP and SPDL+FSP, where the curriculum generator can choose opponents from saved checkpoints. While there are other single-agent curricula RL that regulate difficulty level, such as using regret, we decide to use an approach more relevant to multi-agent domain by including MAESTRO (Samvelyan et al., 2023). MAESTRO represents a stat a state-of-the-art approach optimizing environment-opponent choices in multi-agent domain using regret to regulate difficulty level of the scenarios. Finally, we include FSP as a baseline comparison for simple population-based multi-agent RL. FSP loads environmental parameters by domain randomization. For all algorithms, we use a publicly available implementation of Soft Actor Critic (SAC) (Haarnoja et al., 2018; createmind, 2019) as the base strategy explorer.

## 4.3 EVALUATION AND HYPERPARAMETERS

To evaluate each method, we train with 10 random seeds for Pong and Volley, and 5 for ACM and Race benchmarks. Each seed took 5-15 days to complete. Complex environments like ACM required more time, while computationally intensive algorithms like PONG also had longer run times.

Exploitability analysis isn't feasible in complex environments (Liu et al., 2021), so we measure performance by having agents play against 5 baselines and our GEMS. We play 200 games with random environment parameters per 10x10 pair (5x5 for ACM) of seeds to report the model's performance.

|  | Pong | Volley | ACM |
|---|---|---|---|
| **FSP** | 59±3 : 1±0 : 39±3 | 40±2 : 5±1 : 55±1 | 37±3 : 49±1 : 14±2 |
| **PSRO** | 56±3 : 1±0 : 42±3 | 40±2 : 5±1 : 55±1 | 35±3 : 51±1 : 14±2 |
| **GC+FSP** | 58±4 : 1±0 : 42±4 | 46±2 : 7±2 : 47±1 | 23±5 : 34±3 : 43±3 |
| **SPDL+FSP** | 31±3 : 0±0 : 69±3 | **47±2** : 24±2 : **28±0** | 28±6 : 43±4 : 29±3 |
| **MAESTRO** | 21±3 : 0±0 : 79±3 | 42±2 : 6±1 : 52±0 | 10±3 : 23±3 : 67±2 |
| **GEMS (Ours)** | **72±3** : 3±1 : **25±2** | **48±2** : 24±2 : **28±0** | **41±2** : 52±3 : **8±1** |

Table 1: Mean Win:Tie:Lose Ratio (%) of Algorithms against Baseline Algorithms and Ours. Highest Win Rate and Lowest Lose Rate in Bold

For the training curve, we save model checkpoints at every 5e5 timesteps and had them play Round Robin. Each agent playe 400 games per 10x10 pair (5x5 for ACM) among the seeds for evaluation.

We also conduct an ablation study on our approach to validate our design choices. The ablated versions run with 5 seeds and evaluate against fully trained versions of the baselines for the training curve and overall performance.

While PSRO, GC, and SPDL use additional simulation steps for curriculum generation, we report results based on the steps each agent took during exploration for the ease of comparison. ACM was trained up to 7e6 steps, while the other benchmarks were trained up to 4e6 steps. Implementation details and hyperparameter tuning results can be found in Appendix A.1.

## 5 RESULTS

### 5.1 CROSSPLAY RESULTS

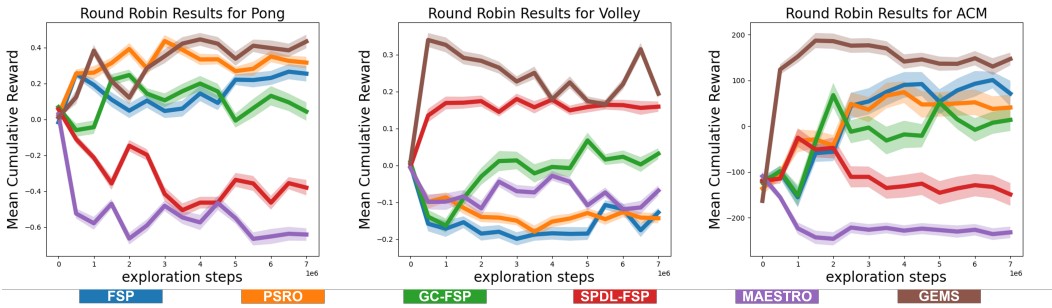

Figure 4: Training Curve with Round Robin Results

Table 1 shows the summary of crossplay results, and the training curves are shown in Figure 4. Our algorithm achieves the highest win rate and lowest lose rate against all benchmarks across all baselines. For full results, refer to Appendix A.6.

Compared to its performance in the Pong benchmark, PSRO does not fare well in relatively more complex benchmarks like Volley or ACM. In Pong, where the players only move in a 1D space, the game is relatively simple, and the game theoretical setup significantly affects the overall performance. However, in the case of Volley, where the players move in a 2D space, and ACM, where the players move in a 3D space, having a curricular setup that assists in learning has a more pronounced effect, leading to PSRO being less effective.

For example, GC+FSP and SPDL+FSP are curricular setups without a equilibrium stability guarantee. While SPDL+FSP outperforms FSP in Volley, which is complex enough to benefit from a curricular setup, SPDL+FSP performs poorly in Pong, where the environment is too simple to benefit from such a setup. GC+FSP does better than SPDL+FSP in PONG thanks to its robustness guarantee, but like SPDL+FSP, it suffers in ACM where the baselines were not designed for multi-agent interactions like our method. We will further emphasize these features in the ablation study.

While MAESTRO performs comparable to FSP in Volley, it does not perform well in other benchmarks. The limitation of relying on random exploration versus a full genetic algorithm for curriculum generation has been reported in GC (Song & Schneider, 2022), and we report that this limitation extends to the multi-agent setup in our ablation study.

## 5.2 Evolution of Generated Curriculum

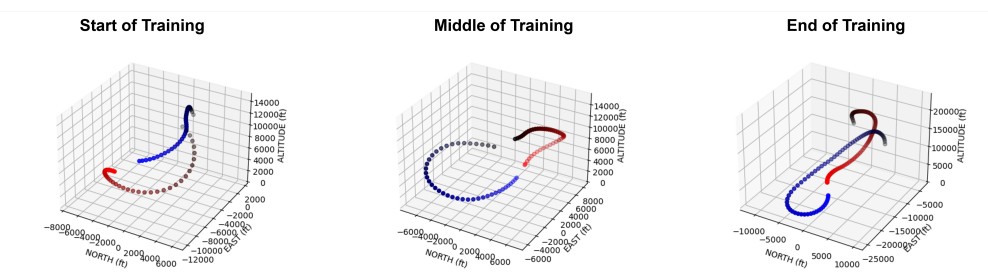

Figure 5: Evolution of Scenarios Generated by Ours. Each dot marks the position of red and blue aircraft at 1-second intervals. The color of the markers starts from black to blue for student aircraft and black to red for the opponent aircraft.

To illustrate the evolution of the curriculum, we present characteristic examples of curricula generated by our approach in Figure 5. We sampled scenarios used to train the algorithm at 1e6, 3e6, and 7e6 steps. At the beginning of the training, when the ego agent is not yet well-trained, our algorithm utilizes Blind Agents to generate opponents with useful demonstrations. In this scenario, it sets the opponent to spawn close and circle around the ego agent, allowing the ego agent to practice basic tracking maneuvers. As the training progresses, our algorithm actively samples scenarios with interesting learning points, such as the scenario in which the agents start flying away from each other. Thanks to these gradual steps, by the end of the training, the agent can explore much more complex situations and opponents. More detailed visualizations, including other baselines, can be found in Figure 7 in the appendix.

## 5.3 Ablation Study

We conducted an ablation study on our proposed method to empirically demonstrate the effects of our design choices. In the **NoRegret** experiment, we excluded the regret term ($\sigma$) when calculating the fitness function to observe how effectively regret regulates the difficulty level of scenarios during training. In the **NoGenetic** experiment, we employed the approach of MAESTRO (Samvelyan et al., 2023) instead of using a genetic algorithm to generate a scenario population. Scenarios were individually selected from a replay buffer, and new scenarios were added through random search, deviating from the batch generation of a genetic algorithm. In the **NoCrossover** experiment, we disabled the crossover function of the genetic algorithm, relying solely on mutation for scenario search. In the **NoBlind** experiment, we disabled the option to use a Blind Agent During Training. Lastly, in the **NoVIC** experiment, we disabled the fitness function from utilizing $G(\pi, \psi)$, focusing solely on maximizing regret.

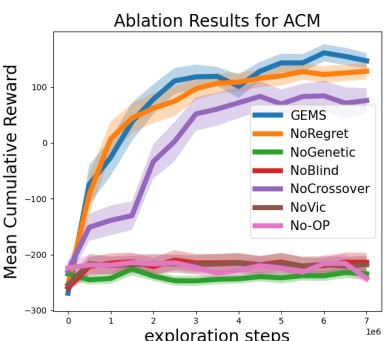

Figure 6: Training Curve For Ablation Study

Overall, we observe that NoRegret initially follows a training curve similar to that of GEMS but reaches a plateau earlier than our proposed GEMS. This suggests that during the early stages of training, the regulation of scenario difficulty is influenced by the improving proficiency of both the opponent agent and the ego agent. However, as the opponent population approaches convergence, regret emerges as a useful means to regulate the curriculum's difficulty level towards approaching

the equilibrium. Nevertheless, the impact of regret on training is not as pronounced as that of other design features in GEMS.

One of the critical features of our curriculum generation is the use of genetic operations. As highlighted in GC (Song & Schneider, 2022), in a single-agent setup, random exploration to find difficult scenarios can be less effective than randomly sampling scenarios, regardless of their quality. The Genetic Algorithm not only excels at searching for scenarios but also promotes similarity between scenarios, aiding in task generation. This effect is observed to extend into the multi-agent domain. Even relying on mutation alone, as demonstrated in NoCrossover, proves to be more effective than the random exploration seen in NoGenetic. The value of the genetic structure is further evident in other benchmarks, as shown by the results of NoGenetic in the Volley and Pong environments.

However, GEMS incorporates certain features not present in GC that prove beneficial in the multi-agent domain, notably the inclusion of the Blind Agent. Without utilizing the Blind Agent, effective exploration becomes challenging for the agent, hindering its ability to escape trivial solutions, especially when the opponent lacks the necessary proficiency to solve the game. As a result of this instability at the outset, NoBlind does not exhibit efficient learning.

Similarly, we observe that the No-OP (no action) approach does not serve as a substitute for the Blind Agent. In the ACM

| ACM | |
|---|---|
| MAESTRO | -226.845±25.332 |
| GEMS | **139.080±32.768** |
| NoRegret | 128.416±14.724 |
| NoGenetic | -234.64±9.595 |
| NoCrossover | 75.808±21.963 |
| NoVIC | -218.267±16.605 |
| NoBlind | -213.751±16.907 |
| No-OP | -243.172±8.521 |
| Volley | |
| MAESTRO | -0.094±0.053 |
| GEMS | **0.194±0.058** |
| NoGenetic | -0.131±0.013 |
| Pong | |
| MAESTRO | -0.578±0.142 |
| GEMS | **0.466±0.121** |
| NoGenetic | -0.552±0.046 |

Table 2: Mean Return of the Ablation Study

environment, an uncontrolled aircraft will inevitably reach a low-energy state, leading to a crash landing. While an aircraft's aerodynamic stability can sustain level flight for a certain duration, this is contingent on adequate airspeed and altitude. However, the aircraft will still gradually lose kinetic and potential energy due to drag. The Blind Agent circumvents this issue by enabling a more sophisticated list of action sequences, optimized by the curriculum generator.

Finally, we observe that NoVic, which corresponds to our multi-agent stability with respect to Nash equilibrium, plays a crucial role in learning. While NoVic still utilizes the regret function to guide exploration, this alone is insufficient to solve the multi-agent exploration problem.

## 6 CONCLUSION AND FUTURE WORK

In this paper, we propose utilizing curricular learning through a genetic algorithm to enhance and stabilize learning in a game-theoretic approach within a multi-agent environment. By employing a genetic algorithm to search for and optimize a population of scenarios for use as a curriculum, we enable an RL agent to reach a better solution faster. Additionally, we introduce the concept of an open-loop agent optimized by the curriculum generator to stabilize the training, especially at the initial stages where the competence of the self-play opponent may not be sufficient to generate valuable training experiences. Through empirical evidence, we demonstrate the effectiveness of our approach against various baselines in several benchmarks. Furthermore, we conduct an ablation study to validate our design choices.

These findings open up several intriguing research directions. One such direction involves scaling up the number of players from 2 to N. Additionally, although our approach is not confined to strictly competitive settings, the empirical studies in this paper focused on such scenarios. It would be interesting to investigate how our approach fares in collaborative-competitive settings. These scenarios could include highway lane merging, where agents are not engaged in a zero-sum game, or team sports, where distributed AI systems must collaborate to compete against an opposing group of AI agents.

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

## A  APPENDIX

### A.1  ENVIRONMENT DETAILS

This section will discuss the details of the benchmark environments used in this paper.

### A.1.1    PONG

**Environment Overview**

Pong is a game similar to the one described in (Brockman et al., 2016). In this game, there are two paddles, each located at the left and right edges of the map. Each agent controls a paddle capable of 1D movement. At the start of the game, a ball spawns at the center of the map with a certain velocity. The ball will bounce if it comes in contact with the top edge, the bottom edge, or any of the paddles. An agent wins the game if the ball passes through the opposite edge of the map.

To play the game, each agent observes the position and velocity of the ball, as well as the location of each paddle. The action space is 1-D. If the value is positive, the agent's paddle will move up at a constant speed, and vice versa if negative. An agent receives a reward of +1 if they win and -1 if they lose. If neither player manages to win the game after 300 steps, it is considered a tie with a reward of 0.

**Environment Encoding**

The original environment calls the random number generator four times to reset a game. Subsequently, we encode the environment with four values. One value determines whether the ego agent will be playing the left or right paddle. The other three control the initial velocity of the ball. One value determines whether the ball will be moving up or down, another controls the magnitude of the ball's velocity vector in the up-down axis, and the last one determines whether the ball will be traveling left or right. The magnitude of the ball's velocity in the left-right axis remains constant throughout the game.

### A.1.2    VOLLEY

**Environment Overview**

Volley is an environment based on the concept presented in (Hardmaru, 2020), where two agents engage in a 2D volleyball game. This presents a more intricate update compared to Pong, as both players can move in a 2D space instead of the 1D movement of paddles. Additionally, the ball's horizontal speed can vary along with its vertical speed. At the start of the game, the ball spawns at the center of the map above the net, which divides the map into two halves. The ball will bounce upon contact with any of the players or any edges except for the bottom. Each of the agents can redirect the ball by hitting it with the avatar they control. The ball follows a simple physics model, being subject to gravity and simple elastic collision mechanics. An agent wins the game by successfully landing the ball on the other side of the map.

To play the game, each agent observes the position and velocity of the ball, the ego agent, and the other agent. Each agent has a 2-D continuous action space, dictating the desired vertical and horizontal velocity of the avatar in the game. While the avatar can move at the desired velocity in the horizontal axis, it can only be launched upward with a desired velocity if it is on the ground. Otherwise, it will move along the surface or be in free fall due to gravity. An agent receives a reward of +1 if they win and -1 if they lose. If neither player manages to win the game after 300 steps, it is considered a tie, and both players receive a reward of 0.

**Environment Encoding**

The Volley environment is expressed in three values. The first two values are continuous (-1,1) and defines the initial velocity of the ball. The third value is discrete 0,1 and defines whether the ego agent is controlling the avatar on the left or the right.

### A.1.3    ACM

**Environment Overview**

ACM is a simulated dogfight environment. This is more complex than the previous 2 benchmarks as agents in ACM will moving in 3D space with realistic physics simulation. At the start of the game, two aircraft spawn in air at certain orientation and a velocity vector with respect to the nose heading. The agent wins the game by pointing its nose at the opponent while avoid getting pointed by the opponents nose. There's also penalty involved with crashing to the ground.

To play the game, each agent observes the position, orientation, and velocity of the ego and the opponent agents. Each agent has 4-D continuous action space with direct control over the aicraft's elevator, aileron, rudder, and throttle. The physics is simulated by JSBSim (Berndt, 2004), a high-fidelity simulator often used in autonomous aircraft research. The aerodynamics model for ACM is based on a Boeing F-15D, capable of flying two times faster than the speed of sound.

To win the game, the agent should have its nose pointed less than 5 degrees off the opponent aircraft while flying at less than 2,000 ft away from the opponent. The agent can also win the game if the opponent aircraft fly below the hard deck of 500 ft. The agent wins the game if either of these conditions are met, and loses if the opponent achieves either of the conditions. If both aircraft achieves either of these conditions at the same time, it's a tie. If none of the conditions are met by neither of the agents for 300 seconds, it's considered a tie. Agent receives penalty of -300 for crashing into the ground, 150 for pointing the nose towards the target, and 100 if the opponent crashed to the ground. To help with training, there's a small dense reward for getting closer to pointing the nose towards the opponent.

**Environment Encoding**

The ACM environment is expressed in 14 continuous values. They define the initial conditions of the game, which are the position, orientation, and airspeed of each aircraft.

## A.2 IMPLEMENTATION DETAILS

This section covers the implementation details regarding the training procedures of each baseline for reproducibility purposes.

### A.2.1 TRAINING HARDWARE

To train the models, we used 88-core Intel Xeon Gold 6238 CPU at 2.10 GHz. Training models took from $5 \sim 15$ days. Lighter benchmarks, like Pong took 5 days, while heavier benchmarks such as ACM took 15 days. This is due to the computation load of the JSBsim which provides high fidelity aerodynamics simulation.

### A.2.2 TUNING RL ALGORITHM

**Tuning Procedures**

To ensure that hyperparameters do not favor one baseline over another, they were tuned for the RL policy explorer in a single-agent setup against hand-coded opponents. For Pong and Volley, the opponents were hand-coded to chase the estimated impact point of the ball, with a PID controller providing the action inputs (Minorsky, 1922). In the case of ACM, a simple PID autopilot was employed to control the opponent aircraft's speed, heading, and altitude. The combination of network structure and hyperparameters that performed the best in these single-agent setups was selected for use in our baseline experiments.

**Network Structure**

To train our RL algorithm, we experimented with various network structures, varying the depth of the hidden layers from 1 to 4 layers and the width from 64 to 512. Our findings revealed that the optimal architecture for all the benchmarks consists of hidden layers composed of two fully-connected layers, each with a size of 256. We utilized ReLU activation between the hidden layers and concluded with the last layer having an output dimension equivalent to the action dimension, followed by tanh activation.

### A.2.3 RL HYPERPARAMETERS

We conducted limited grid search to hyperparameters for our RL algorithm. For learning rate we tried values of {3e-5, 1e-4, 3e-4, 1e-3}. For initial exploration steps, we tried 0, 1000, 10000. For discount ratio $\gamma$, we tried {0.95, 0.98, 0.999}. For batch size, we tried {64, 256, 512}. For update every, we tried {1, 32, 100}. For Replay size, we tried {1e6, 3e6}. As a balance between perfomance and computational cost, we ran each seeds on 3 seeds. Table 3 shows the selected hyperparameters we used for the main experiments.

|                | Pong  | Volley | ACM   |
|----------------|-------|--------|-------|
| **Start Steps**    | 10000 | 10000  | 10000 |
| **Learning Rate**  | 3e-4  | 3e-4   | 3e-4  |
| $\gamma$           | 0.98  | 0.98   | 0.98  |
| $\alpha$           | auto  | auto   | auto  |
| **Batch Size**     | 256   | 256    | 512   |
| **Replay Size**    | 1e6   | 1e6    | 1e6   |
| **Update Every**   | 1     | 1      | 1     |

Table 3: Selected Hyperparameters for RL Algorithm

|                                    | Pong  | Volley | ACM   |
|------------------------------------|-------|--------|-------|
| **PSF**                            |       |        |       |
| **Checkpoint Interval**            | 10000 | 10000  | 10000 |
|                                    |       |        |       |
| **PSRO**                           |       |        |       |
| **Match Times**                    | 30    | 30     | 10    |
|                                    |       |        |       |
| **GC**                             |       |        |       |
| **Evaluation Set Size**            | 200   | 200    | 300   |
| **Curriculum Size**                | 200   | 200    | 300   |
|                                    |       |        |       |
| **SPDL**                           |       |        |       |
| **Penalty Proportion**             | 1     | 1      | 0.1   |
| **Offset**                         | 20    | 20     | 20    |
|                                    |       |        |       |
| **MAESTRO**                        |       |        |       |
| **Co-Player Exploration Coefficient** | 0.1   | 0.1    | 0.1   |
| **Curriculum Buffer Size**         | 1000  | 1000   | 1000  |

Table 4: Selected Hyperparameters for Baselines

### A.3 BASELINE HYPERPARAMETER TUNING

To tune the hyperparameters, we trained the models using the following settings. Since it is challenging to tune the hyperparameters by conducting a round-robin across all hyperparameter settings used for different baselines, each setting was evaluated against the same hand-crafted opponents used for tuning the RL hyperparameters. Striking a balance between performance and computational cost, we ran each configuration with 3 seeds. Table 4 presents the selected hyperparameters used for the main experiments.

We tested the following hyperparameter values in a limited grid search. For adding checkpoints, we tested saving a copy of the agents' policies at every $\{10000, 30000\}$ exploration step. We used the same interval of steps as the interval in which curriculum is generated for the algorithms that generate one at every epoch. For measuring performance between each checkpoint on random environment parameters to approximate the Nash Equilibrium in PSRO, we tried $\{10,30\}$ evaluations per pair. As for the hyperparameters of the GC, we tried sizes of $\{200,300\}$ for the size of the population of scenarios evaluated for whether being solved or not solved. We also tried sizes of $\{200,300\}$ for the sizes of the generated curriculum and mutation rates of $\{0.1,1\}$. For SPDL, we tried values of $\{0,10,20\}$ for offset and $\{0.1,0.3,1\}$ for penalty proportion. For MAESTRO, we tried co-player exploration coefficients of $\{0.05,0.1\}$ and curriculum buffer sizes of $\{500,1000\}$. For GEMS, we copied the hyperparameters from the GC in terms of the size of the curriculum and mutation rate. This was done to make comparisons between different algorithms easier by removing the effect of different hyperparameters during genetic operations.

### A.4 CROSSOVER

#### A.4.1 CROSSOVER OPERATIONS

For a sampled parent scenarios $\mathbf{m}, \mathbf{n}$, the corresponding encoding would look like;

$$\mathbf{m} = \{i_\pi, \pi_\varnothing, \psi\} = \{m_0, m_1\} = \{m_{0,0}, m_{0,1}...m_{0,x}, m_{1,0}, m_{1,1}...m_{1,y}\} \tag{3}$$

$$\mathbf{n} = \{i_\pi, \pi_\varnothing, \psi\} = \{n_0, n_1\} = \{n_{0,0}, n_{0,1}...n_{0,x}, n_{1,0}, n_{1,1}...n_{1,z}\} \tag{4}$$

$\mathbf{m}_o$ encodes the choice of opponent to load $i_\pi$ and the environment parameters $\psi$ while $\mathbf{m}_o$ represents the encoding of the Blind Agent $\pi_\varnothing$ .In this section, $x$ corresponds to the length of environment encoding for $\mathbf{m}$. $y$ and $z$ in this section corresponds to the length of Blind Agent $\pi_\varnothing$ encoding for $\mathbf{m}$ and $\mathbf{n}$. If using the same environment, $x$ will not be different between the scenarios whereas $y$ and $z$ will vary depending on the encoded environment.

A crossover is performed by swamping a section from one parent with a section from another parent. In this set, this would be done by finding three split points. Using a uniform distribution, we sample three integer values $\eta_0 \in \{0, 1, 2, ..x\}, \eta_1 \in \{0, 1, 2, ..y\}, \eta_2 \in \{0, 1, 2, ..z\}$

From this, we would generate two child scenarios $\mathbf{p}, \mathbf{q}$ as follows;

$$\mathbf{p} = \{m_{0,0}...m_{0,\eta_0}, n_{0,\eta_0+1}...n_{0,x}, m_{1,0}...m_{1,eta_1}, n_{1,eta_2+1}...n_{1,z}\} \tag{5}$$

$$\mathbf{q} = \{n_{0,0}...n_{0,\eta_0}, m_{0,\eta_0+1}...m_{0,x}, n_{1,0}...n_{1,eta_2}, m_{1,eta_1+1}...m_{1,y}\} \tag{6}$$

For the special cases such as $\eta_0 = 0, x$ or $\eta_1 = 0, y$, or $\eta_2 = 0, z$, we would be inherting the segment without dividing it. For example, if $\eta_0 = 0$, then $\mathbf{p_0} = \mathbf{n_0}, \mathbf{q_0} = \mathbf{m_0}$ and so on.

To perform a mutation on a sampled scenario, we would first perform a crossover between the scenario and a randomly generated scenario and return one of the child scenarios as the mutated scenario. The mutation rate controls the probability of each scenario undergoing mutation.

### A.5 SAMPLING PARENTS

In this paper, we have defined the probability of sampling scenarios as parents as the following;

$$p(\xi) \propto \delta(\xi)(1 - G(\pi, \xi)) \tag{7}$$

In practice, we would be using the normalized form as follows;

$$p(\xi_i) = \frac{\delta(\xi_i)(1 - G(\pi_i, \xi_i))}{\Sigma\delta(\xi)(1 - G(\pi, \xi))} \tag{8}$$

If $\sum \delta(\xi)0$, we would simple use regret-only version as follows;

$$p(\xi_i) = \frac{(1 - G(\pi_i, \xi_i))}{\Sigma(1 - G(\pi, \xi))} \tag{9}$$

### A.6 FULL PERFORMANCE RESULTS OF THE TRAINED ALGORITHMS

In this section, we include the full results of our experiments. Table 5 shows the mean return of each algorithm against each other, while Table 6 includes the detailed results of each pair.

### A.7 EVOLUTION OF CURRICULUM

Figure 7 includes characteristic examples of the scenarios generated and used as curriculum for each algorithms.

|  | Pong | Volley | ACM |
|---|---|---|---|
| **FSP** | $0.197 \pm 0.158$ | $-0.143 \pm 0.061$ | $94.498 \pm 49.686$ |
| **PSRO** | $0.141 \pm 0.154$ | $-0.145 \pm 0.058$ | $83.274 \pm 51.499$ |
| **GC+FSP** | $0.159 \pm 0.179$ | $-0.004 \pm 0.07$ | $-124.304 \pm 44.356$ |
| **SPDL+FSP** | $-0.386 \pm 0.144$ | $0.192 \pm 0.058$ | $-97.987 \pm 51.969$ |
| **MAESTRO** | $-0.578 \pm 0.142$ | $-0.094 \pm 0.053$ | $-226.845 \pm 25.332$ |
| **GEMS (Ours)** | $\mathbf{0.466 \pm 0.121}$ | $\mathbf{0.194 \pm 0.058}$ | $\mathbf{139.080 \pm 32.768}$ |

Table 5: Mean Return of Algorithms against Baseline Algorithms and Ours. Highest Mean Return in Bold

Table 6: Detailed Win:Tie:Lose Ratio(%) of Each Algorithm Against Others

| Agent | vs FSP | vs PSRO | vs GC+FSP | vs SPDL+FSP | vs MAESTRO | vs GEMS (Ours) |
|---|---|---|---|---|---|---|
| **Pong** | | | | | | |
| FSP | 49±1:2±0:49±1 | | | | | |
| PSRO | 42±1:1±0:57±1 | 49±1:3±0:49±0 | | | | |
| GC+FSP | 48±1:1±0:51±1 | 22±1:1±0:78±1 | 50±1:0±0:50±1 | | | |
| SPDL+FSP | 22±0:0±0:78±0 | 19±0:0±0:80±0 | 40±0:0±0:60±0 | 50±0:0±0:50±0 | | |
| MAESTRO | 9±0:0±0:91±0 | 13±1:0±0:87±1 | 8±0:0±0:92±0 | 89±0:1±0:11±0 | 50±0:0±0:50±0 | |
| GEMS (Ours) | 66±1:4±0:30±0 | 63±1:4±0:33±1 | 73±1:2±0:25±1 | 94±0:0±0:6±0 | 47±1:7±0:47±1 | 50±0:0±0:50±0 |
| **Volley** | | | | | | |
| FSP | 50±0:0±0:50±0 | | | | | |
| PSRO | 50±0:0±0:50±0 | 50±0:0±0:50±0 | | | | |
| GC+FSP | 58±0:0±0:42±0 | 59±0:0±0:41±0 | 49±0:2±0:49±0 | | | |
| SPDL+FSP | 59±0:16±0:25±0 | 59±0:15±0:26±0 | 51±0:21±0:29±0 | 30±0:41±0:30±0 | | |
| MAESTRO | 51±0:0±0:49±0 | 52±0:0±0:48±0 | 46±0:0±0:54±0 | 29±0:15±0:56±0 | 50±0:0±0:50±0 | |
| GEMS (Ours) | 61±0:13±0:26±0 | 58±0:17±0:25±0 | 52±0:19±0:29±0 | 31±0:37±0:31±0 | 54±0:20±0:27±0 | 31±0:38±0:31±0 |
| **ACM** | | | | | | |
| FSP | 25±0:49±0:25±0 | | | | | |
| PSRO | 20±0:54±0:26±0 | 22±0:55±0:22±0 | | | | |
| GC+FSP | 3±0:48±0:49±0 | 3±0:50±0:47±1 | 38±2:23±1:38±2 | | | |
| SPDL+FSP | 2±0:48±0:50±0 | 2±0:49±0:49±0 | 63±2:23±1:15±1 | 8±0:84±0:8±0 | | |
| MAESTRO | 1±0:39±0:60±0 | 1±0:39±0:60±0 | 10±1:13±1:78±1 | 2±0:5±0:93±0 | 47±1:6±0:47±1 | |
| GEMS (Ours) | 32±0:56±0:14±0 | 35±0:55±0:12±0 | 50±0:48±0:2±0 | 48±0:50±0:1±0 | 62±0:37±0:1±0 | 18±0:64±1:18±0 |

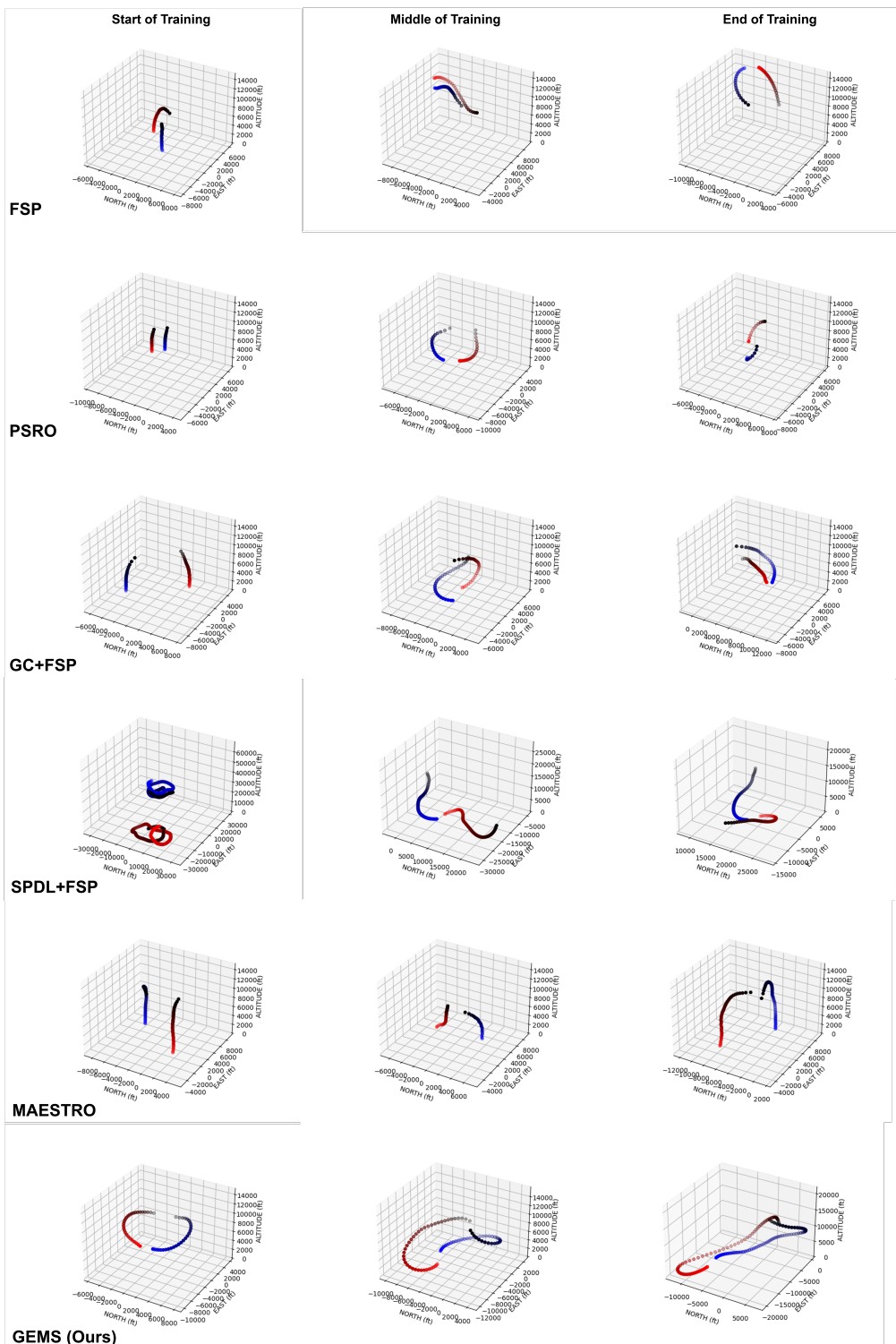

Figure 7: Each dot marks the position of the red and blue aircraft at 1-second intervals. The color of the markers transitions from black to blue for the student aircraft and from black to red for the opponent aircraft. It's important to note that at the start of the training, only our algorithm is capable of presenting scenarios and opponents that provide interesting data points instead of simply crashing to the ground. While SPDL+FSP somewhat succeeds in finding scenarios that do not end in a crash, it optimizes scenarios without considering opponent policy and simply finds trivial cases where agents are flying in circles.

