# OpenReview forum: "Genetic Algorithm for Curriculum Generation in Multi-Agent Reinforcement Learning"
_ICLR.cc/2024/Conference — Submitted to ICLR 2024_

### Official Review · Reviewer_63gd · 2023-10-14

**Soundness:** 3 good
**Presentation:** 2 fair
**Contribution:** 2 fair
**Rating:** 5
**Confidence:** 3

**Summary:**

The authors propose a Curriculum Learning approach for Adversarial 1x1 RL. A Genetic Algorithm generates tasks (i.e., samples opponents) that the agent can use to train, with the purpose to improve it's general performance against a general opponent.

**Strengths:**

- Work is relevant to ICLR and adherent to the recent body of research on RL.
- The idea of generating a Curriculum of tasks to learn how to play against a population of opponents is reasonable and has good applicability to practical applications.
- Authors present evaluation domains of high complexity (as far as 1x1 domains go).

**Weaknesses:**

- The approach is very unclear from the general modeling to how the experiments were performed. It's very hard to judge if the modeling is realistic and the experiments were carried out in a fair way. I will break down the main sources of confusion below:

 -- It is not clear to me what is the "population" of agents that the ego agent has to beat on equation (2). In a practical application, is the agent expected to have to beat an unknown population of agents? If the population is unknown (which sounds to me a more realistic modeling), how is the agent able to generate the strategies it plays against when the curriculum is created? If the population of opponents is known beforehand (which sounds very unrealistic for most applications), it explains how the agent can generate the curriculum.

 -- I don't quite understand what is a "solution" for the GA so that crossover and mutation is applied. The agent can realistically pick which opponent it is going to play against for a particular episode, what else composes the "curriculum task:? how are those combined/mutated. Those are key elements of the proposed algorithm and are not clearly explained in the main text of the paper. In special, I cannot understand what the agent has the power to manipulate to generate a new "curriculum task".

 -- The "blind agent" sounds to be a very ad hoc addition to the algorithm. How should this blind agent be developed for a new domain? Is there an automated way of generating it? I assume just generating an agent with random actions would have no effect given that for any reasonably-complex task a random agent would be very easily beaten.

 -- It's not clear what are the 5 "baseline opponents" used to test the algorithms in the experimental evaluation. Are the agents aware of those opponents and able to use them for training for as long as they have remaining training steps? or are they held out just for generating metrics. An interesting way of validating the algorithms would be to have a number of high-performance strategies they can manipulate to play against during training, and a number of unknown strategies that are never seen by the agents and only used for calculating the performance metrics. In this way you would actually be evaluating how general is the learned strategy.

 -- How strong are the strategies that the agents had to beat in the experiments, and how were them learned? In the evaluation, there should be at least one adaptive agent that is given time to learn how to "hack" the ego agent strategy.

- THe literature review was very limited on the transfer of information and curriclum in multi-agent systems. I was surprised that the authors did not mention the first survey to suggest to develop multi-agent curricula of evolving strategies:

Silva, Felipe Leno Da, and Anna Helena Reali Costa. "A survey on transfer learning for multiagent reinforcement learning systems." Journal of Artificial Intelligence Research 64 (2019): 645-703.

Also, while the agents knowing and controlling the opponent strategies sounds unrealistic, they could easily model then and build Curricula based on modeled versions of those opponents:

Stefano V. Albrecht, Peter Stone, Autonomous agents modelling other agents: A comprehensive survey and open problems, Artificial Intelligence, Volume 258, 2018.

One issue that some Curriculum Learning approaches observed is that it is hard to figure out how many tasks have to be generated. One popular approach executes a random walk based on the task transferability to prune huge Curricula:

Silva, F. L. D., & Costa, A. H. R. (2018, July). Object-oriented curriculum generation for reinforcement learning. In Proceedings of the 17th international conference on autonomous agents and multiagent systems (pp. 1026-1034).

**Questions:**

1) How much knowledge do the agents have about the real "opponent population".
2) What exactly is manipulated by the GA to generate new tasks?
3) Why is the "blind agent" development not added to the algorithm?
4) Where do the strategy from the opponents in the experiments come from? Are there adaptive agents as opponents?

---

> ### Author Response · Authors · 2023-11-21
>
> We thank the reviewer for the helpful feedback! Here's part 1 of our response.
>
> Point 1: Since we are using a minmax formulation for equation 2, Ξ is the curriculum of scenarios (opponent policy-environment parameter pairs) that the ego agent π has difficulty in solving. Therefore, the opponent policy would be the one that minimizes the zero-sum return of the ego agent π for the given environment parameter ψ
>
> Point 2: A curriculum Ξ consists of scenarios ξ. Scenario defines opponent policy π_opp and environment parameters ψ. Using a genetic algorithm, we would migrate the distribution of scenarios in a curriculum that minimizes the zero-sum outcome for the ego agent and maximizes regret. This optimization of the curriculum is done by a genetic algorithm without expert supervision or fine-tuning.
>
> Point 3: We used simple crossover and mutation operations commonly used for data sequences. For crossover, we start by choosing two parent scenarios based on fitness function (the optimization goal explained above). We then chose a random segment from one of the scenario encodings and swapped it with a random segment from the other scenario encoding. In the case of mutation, we randomly choose a parent and perform a crossover with a randomly initialized scenario.
>
> We also note that engineering effort on the scenario encoding is minimal, as this is simply a string of integers and floats in an order. The first part of this sequence is read by the environment code to load the environment. The remaining part of the sequence is used by the policy loader where it will load the corresponding RL or Blind agent to act as the opponent policy.
>
> Point 4: We believe that adapting Blind agents to a new domain is quite easy since we did not use expert supervision or human fine-tuning to initialize the population. This is the code we used to initialize the Blind Agents for our experiments.
>
>     # max_ep_len : Maximum length per episode of the given environment
>     # action_dim : dimension of the action space.
>     import numpy as np
>     def initialize_blind_agent(max_ep_len,action_dim):
>         timestamp_count = np.random.random.randint(1,max_ep_len)
>
>         genome = []
>         for _ in range(timestamp_count):
>             actionCallout = []
>             actionCallout.append(np.random.randint(max_ep_len)) #Time
>             actionCallout.append(np.random.uniform(low=-1,high=1,size=action_dim)) #Action
>             genome.append(actionCallout)
>
>         return genome
>
> During training, we record the outcome of each blind agent action sequence in the population. Using mutation and crossover, we use this data to genetically optimize the generate the population of blind agents based on the fitness function p(ξ) as explained in Section 3.3
>
> To the best of our knowledge, we are the first paper to train a competitive multiagent RL policy with a genetically optimized opponent. In the existing literature, opponents are either provided by expert supervision (log replay data or behavior cloning based on expert supervision) or via self-play. While expert supervision can help an ego agent escape out of local minima at the start, expert supervision is costly. Self-play, on the other hand, does not require self-play but has a tendency to overfit to a local minimum. Our approach uses a genetic algorithm to provide an opponent that helps the ego agent from overfitting while not requiring expert supervision.
>
> The key feature of a blind agent is that it is a genetically optimized opponent policy that is blind to adversarial noise. If we use random policy instead of genetically optimized policy, it works in simple environments such as Pong. However, as the reviewer rightfully pointed out, in a complex environment like the ACM, a random policy would be too easy to beat as it would just crash to the ground. The training suffered due to the long interim phase where the RL policy is winning random policies by staying in the air longer than the random policy that keeps crashing to the ground but has not yet learned useful tricks that facing another RL agent would be interesting. Using genetically optimized blind agents smooths the transition between training against open-loop opponents to avoid overfitting and training against closed-loop RL agents to know what to do against challenging policies.
>
> Point 5: In our experiments, we trained 5 baselines (FSP, PSRO, GC+FSP, SPDL+FSP, MAESTRO) and our proposed algorithm (GEMS). These 6 algorithms played against each other in a 6x6 pair for the evaluations. None of the 6 algorithms saw each other during training. For example, GEMS would have seen a GEMS agent during training, but would not have seen FSP, PSRO, GC+FSP, SPDL+FSP, nor MAESTRO agent during training.
>
> Point 6: We thank the reviewer for making suggestions on papers to cite. We agree that these papers are relevant to and share interesting insights about our approach and will include the citations in our final draft.

---

> > ### Author Response · Authors · 2023-11-21
> >
> > We thank the reviewer for the helpful feedback! Here's part 2 of our response.
> >
> > Question 1: The agent has no knowledge about the real opponent population it’s going to face in the test time. One of the main motivations for using a zero-sum formulation is that by using such formulation, we can turn the multi-agent problem into minmax problem, reducing the area the curriculum has to search.
> >
> > Question 2: The Genetic Algorithm has control over the environmental parameters and the choice of opponents (RL agents loaded from previous checkpoints or Blind Agents genetically optimized).
> >
> > Question 3: Blind Agents are not developed by humans. They are optimized by a genetic algorithm without expert supervision during training time.
> >
> > Question 4: During training time, opponents consist of 1) RL agents loaded from previous checkpoints of ego agents or 2) Blind Agents genetically optimized by GEMS. During test time, we use 5 baselines and our proposed algorithm, GEMS to play against. Note that none of the 6 algorithms ever saw each other during training.
> >
> > We again thank the reviewer for the helpful feedback that will surely make our paper stronger! We’ll make sure to include the feedback to our final draft.

---

> > > ### Comment · Reviewer_63gd · 2023-11-22
> > >
> > > I am maling some additional comments below, but in general I maintain my evaluation that while the approach has value, the paper is unclear and cam be significantly improved. Therefore I keep my score.
> > >
> > > Point 1: the answer still doesn't clarify if you have to *compute* the strategy that works best against the ego agent or if you just choose one of a few options, and where those policies come from. This shoukd be made more clear in the manuscript.
> > > Points 2 and 3: the scenario encodings are still not clear to me, I suggest detailing in the paper how thode encodings are generated in general for a new scenario and giving some examples.
> > > Point 4: if I understand correctly the blind agent will simply apply random actions. For many domains that would result in an agent that has a terrible performance and completely trivial to beat, which means beating it has not much value apart from saying that the agent policy is better than random. In this situation I expect you might run into issues

---

> ### Author Response · Authors · 2023-11-22
>
> Point 1: Using the minmax formulation, what we are trying to optimize with our curriculum generator is to find the scenario (the combination of policies and environmental parameters) that performs the best against the ego agent. The population of policies comes from the co-player population which is compiled by periodically saving the checkpoints of the ego agent during training, as shown in Figure 1. The role of the curriculum generator is performed by a genetic algorithm. Which uses the fitness function derived from the minmax formulation to evolve the scenario encodings (which encodes environmental parameter, index number denoting which checkpoint to load as policy or a string defining blind agent) that will be challenging to the ego agent.
>
> Points 2 and 3: Certainly! The Section 3 describes how the scenarios are initialized and how the scenarios are generated for the subsequent curriculum using a genetic algorithm. However, we can describe again how the scenarios are initialized at the training and how a new curriculum is generated during the training.
>
> At the start of the training, scenarios are randomly initialized. In the case of the Volley, the environment code needs three numbers to initialize an environment. 2 floats describing the initial velocity of the ball and 1 boolean value to denote which side (left or right) will the ego agent be playing.
>
> Therefore, the scenario initialization code for the Volley is defined as follows;
>
>     # max_ep_len : Maximum length per episode of the given environment
>     # action_dim : dimension of the action space.
>     # num_policies : #Number of policies available
>     import numpy as np
>     def initialize_scenario(max_ep_len,action_dim)
>         scenario = []
>         scenario.append(np.random.uniform()) #X velocity of the ball
>         scenario.append(np.random.uniform()) #Y Velocity of the ball
>         scenario.append(np.random.randint(1)) #Which side is the ego agent playing?
>         scenario.append(np.random.randint(num_policies+1)) #Which checkpoint is being used as the opponent policy? If this value is zero, then use the blind agent as the opponent
>         scenario.append(initialize_blind_agent()) # Adds the string of instructions for a blind agent.
>         return scenario
>
> After the first curriculum is initialized with the scenario initialization function, the subsequent curriculums are generated by the genetic algorithm, which will perform crossover and mutation to generate more scenarios similar to ones that performed well against the opponent.
>
> Point 4: While the blind agent is first initialized with random actions, they don’t just end up applying random actions throughout the training. Because of the genetic algorithm’s optimization function used to generate the subsequent curriculums and scenarios, only the action sequences that resulted in a good performance against the ego agent will be kept and continue to evolve. This allows the curriculum generator to explore and generate more competitive blind as the training progresses. For example, in the ACM benchmark, our curriculum generator in the first few epochs quickly removed all blind agents that resulted in the opponent agent crashing to the ground as this gives an easy victory to the ego agent.
>
> We hope that we have addressed the questions raised and will update our explanations accordingly for the final draft. Please feel free to let us know if you have any questions or find points not well addressed by our response!

---

### Official Review · Reviewer_B2gt · 2023-10-30

**Soundness:** 3 good
**Presentation:** 3 good
**Contribution:** 2 fair
**Rating:** 5
**Confidence:** 4

**Summary:**

This paper presents GEMS that addresses unstable training of existing work towards Nash caused by complex multi-agent interactions. Specifically, GEMS applies 1) population-wide genetic operations, 2) regret to evaluate the difficulty of a generated scenario, and 3) continuously optimized opponents, environment parameters, and the blind agent. Experimental evaluations in Pong, Volley, and ACM show the effectiveness of GEMS compared to competing baselines.

**Strengths:**

1. The paper is well-written and addresses the important challenge of learning in difficult multi-agent domains.
2. The paper conducts extensive crossplay experiments (e.g., Table 6) and shows the positive results of GEMS.

**Weaknesses:**

1. GEMS would have a limited novelty with respect to prior work: the population-wide genetic operations (e.g., crossover) are studied in GC and the use of regret for curriculum learning is studied in MAESTRO. As such, GEMS could be viewed as combining two papers with the addition of the blind agent.
2. Related to #1, the ablation study in Section 5.3 shows that the use of the blind agent is an important factor in GEMS. However, the initial behavior of the blind agent is manually set by a human (e.g., the blind agent is set to circle around the ego agent (Section 5.2)). As such, GEMS would require a human to manually tune the initial blind agent.
3. The paper states that "Over time, those opponents will evolve toward the Nash equilibrium". However, there are no theoretical analyses, and it is unclear from empirical evaluations that GEMS converges to Nash equilibrium.
4. It is unclear how to scale GEMS to more complex multi-agent domains that involve images as agent inputs. Also, the problem statement (Equation 1) in Section 3.1 is only with respect to the ego agent, so it is unclear when there is more than one ego agent in settings.

**Questions:**

1. I hope to ask the authors' responses to my concerns about the limited novelty, human requirement, Nash justification, and limited scalability (please refer to the weaknesses section for details).
2. In Figure 2, it is unclear what each color represents. Adding a legend in the figure could help.
3. Missing reference: "[?]" in Section 3.3
4. Typo: "a stat a state-of-the-art approach" in Section 4.2

---

> ### Author Response · Authors · 2023-11-21
>
> We thank the reviewer for insightful and helpful comments! Here’s our response to questions and weaknesses raised in the reviewing process.
>
> Weakness 1&2:
>
> The initial behavior of the blind agent is not manually set by a human. The initial behavior of the blind agent is sampled using a uniform random function as the following;
>
>     # max_ep_len : Maximum length per episode of the given environment
>     # action_dim : dimension of the action space.
>     import numpy as np
>     def initialize_blind_agent(max_ep_len,action_dim):
>         timestamp_count = np.random.random.randint(1,max_ep_len)
>
>         genome = []
>         for _ in range(timestamp_count):
>             actionCallout = []
>             actionCallout.append(np.random.randint(max_ep_len)) #Time
>             actionCallout.append(np.random.uniform(low=-1,high=1,size=action_dim)) #Action
>             genome.append(actionCallout)
>
>         return genome
>
> To the best of our knowledge, we are the first paper to train a competitive multiagent RL policy with a genetically optimized opponent. In the existing literature, opponents are either provided by expert supervision (log replay data or behavior cloning based on expert supervision) or via self-play. While expert supervision can help an ego agent escape out of local minima at the start, expert supervision is costly. Self-play, on the other hand, does not require self-play but has a tendency to overfit to a local minimum. Our approach uses a genetic algorithm to provide an opponent that helps the ego agent from overfitting while not requiring expert supervision.
>
> The image in the section 5.2 shows a good example of how genetic algorithm can be used to provide a good training scenario without expert supervision. From the Figure 5, we can see that the genetic algorithm in less than 10 epochs has already found a good starting position and flight input commands resulting in opponent aircraft circling around to give the ego agent good time to practice chasing. This is noteworthy considering that a genetic algorithm has found this training scenario early in the training without any expert supervision.
>
> Weakness 3:
>
> We think that a provable guarantee of convergence to Nash Equilibrium is a topic that should be covered in future works. However, it should be noted that a provable guarantee of convergence is a difficult problem not well addressed in existing works as well. One such case would be MAESTRO (Samvelyan et al, 2023) which describes ‘identifying conditions whereby such an algorithm can provably converge to Nash equilibrium’ as a possible future work.
>
> However, following a curricular approach, we think that curriculum RL has the potential to help multiagent RL policies reach a Nash Equilibrium. In the PSRO framework of multiagent reinforcement learning (RL), the focus is on finding an approximate Nash Equilibrium solution to train against. However, in the case of the single agent RL, it has been shown that directly training against the goal task can be difficult and unachievable, and breaking down the learning task with a curriculum can allow an agent to achieve a better steady-state solution faster. We think that curricular multiagent approaches such as MAESTRO and ours has the potential to improve training performance by providing a curriculum to help ego agent learn competitive skills. However, we acknowledge that proof of convergence is an important research question that should be addressed in future works.
>
> Weakness 4:
>
> We agree that it would be interesting to see how well GEMS scales to complex domains with images as inputs and more than one ego agent. We are currently running additional experiments where 1) we attempt to solve Volley with images as inputs and 2) we play a game of tag with 4, 16, and 32 players in total. While we’ll have to wait until tomorrow to complete the training, the current training progress looks promising. We’ll post the results when they are ready before the end of the rebuttal period.
>
> Question 2:
>
> Each color represents the scenario encodings originally came from. So in the case of crossover, we mix two parents (green and purple) to make two children scenarios. We will edit the figures to be clearer in the final draft.
>
> Question 3 & 4:
>
> The missing reference in Section 3.3 is the paper, “Replay-guided Adversarial Environment Design” (Jiang et al, 2021). We thank the reviewer for pointing this out and will fix the issue, along with other typos, for our final draft.

---

> > ### Comment · Reviewer_B2gt · 2023-11-22
> > **Response to Rebuttal**
> >
> > I appreciate the authors for providing a detailed response to my feedback.
> >
> > Weaknesses 1 and 2: Thank you for correcting my misunderstanding about the initial behavior of the blind agent.
> >
> > Weakness 3: If convergence to Nash is not guaranteed or cannot be justified in the current paper, it might be helpful to emphasize less or remove the statements about Nash in paper writing.
> >
> > Weakness 4: Thank you. I look forward to seeing new results.

---

> ### Author Response · Authors · 2023-11-22
>
> We thank the reviewer for taking time and effort to give us additional feedback!
>
> We are happy to hear that our explanations has made our approach clearer. We'll make sure to include this in our final draft.
>
> We also agree with the reviewer's comment regarding Nash Guarantee. We will make this limitation and assumptions clearer in the final draft. We'll also include this weakness as a future research topic.
>
> Some of our reviewers wondered how scalable our algorithm scales with respect to the number of players in a game. We therefore ran the SimpleTag environment in the PettingZoo (Terry et al, 2021) benchmark.
>
> In this predator-prey environment. There are n-players taking the two teams. In the ‘Good’ team, players get a negative reward (-10) every time they run into members of the ‘Bad’ team. The members of the ‘Bad’ team are slightly slower than the ‘Good’ team and get a positive reward (+10) every time they run into the good team. To make the environment more interesting, there are two large obstacles occupying the map, forcing the agents to maneuver around the obstacle. Each agent observes its own speed, location, the location of the obstacle, the location of the other agents, and their velocities. To help with learning, there’s a sparse dense reward function associated with chasing and evading.
>
> In the ST4 benchmark, there are a total of 4 players in the game, 2 playing as ‘Good’ and 2 playing as ‘Bad’. Trained for 4e5 steps with 5 seeds, the performance of trained algorithms against each other is as follows;
>
>
> ALGORITHM  |FSP                 |PSRO                |GC+FSP              |SPDL+FSP            |MAESTRO             |GEMS
>
> MEAN REWARD|-14.516±0.517       |-20.025±0.746       |-26.054±1.232       |-27.13±0.586        |-22.722±0.765       |-12.624±0.673
>
>
> In the ST8 benchmark, there are a total of 16 players in the game, 8 playing as ‘Good’ and 8 playing as ‘Bad’. Trained for 2e6 steps and with 5 seeds, the performance of trained algorithms against each other is as follows,
>
>
> ALGORITHM  |FSP                 |PSRO                |GC+FSP              |SPDL+FSP            |MAESTRO             |GEMS
>
> MEAN REWARD|-1.353±1.681        |-6.472±1.533        |-0.002±1.286        |0.942±2.202         |4.193±1.515         |6.403±1.52
>
>
> In all experiments regarding scalability towards multiple players, our agent has successfully scaled from 1v1 to 8v8 cases and has outperformed all baselines. For our final draft, we'll run 5 additional seeds to report the performance on 10 seeds.
>
> In regards to images as inputs, we note that our algorithm is not specific to the type of input, whether it’s an small 1D array containing concentrated information or an 2D array containing raw images. While scaling to image based input should be straightforward, we were not able to finish the training within the rebuttal period.

---

### Official Review · Reviewer_wvYt · 2023-10-30

**Soundness:** 3 good
**Presentation:** 3 good
**Contribution:** 2 fair
**Rating:** 5
**Confidence:** 3

**Summary:**

This paper introduces a population-wide genetic algorithm that aims to optimise agents in competitive multi-player settings. In particular, it utilises curriculum learning techniques to generate both scenarios and opponents in order to improve the training process of an agent.

**Strengths:**

- In general, the paper is fairly well-written and the method is described adequately.
- Whilst the curriculum learning aspect of the paper is not necessarily truly novel, combining it across both the environment and the opponent players I believe is novel. Frameworks like PSRO only perform curriculum learning by selecting appropriate opponents, but ignore the underlying environment, whilst single-agent curriculum learning approaches obviously only generate curriculums on the environment. Therefore, in my opinion performing the curriculum learning is an obvious step but is mostly original work.
- The experimental results suggest that this is a fruitful approach on all of the environments.

**Weaknesses:**

- Whilst I understand the intentions by the blind agent, from what is provided in the main text I am struggling a bit in figuring out the implementation. For example, how is the action selected? In addition, why this approach in particular? Could we not get a similar result by e.g. applying noise to the policy when a blind agent is chosen to be used? What about just a random policy?
- The curriculum learning aspects of the environments evaluated in the paper are very limited. For example, in Volley the initial velocity of the ball does provide an obvious form of curriculum (start slow and speed up as agent gets better), however this is incredibly simple and does not provide really provide much signal for the agent. I think it would be much more interesting if this framework was tested in environments where there exists a lot of potential complexity in the underlying environment, that can also be stripped down to a simple version for the curriculum (e.g. single-agent MiniGrid has much more scope for going from basic to complex unlike these environments, in my opinion)

**Questions:**

I would appreciate if the authors could address the concerns I highlighted in the weaknesses section. Primarily:

1) Implementation details of the Blind Agent and why this specific design was selected

2) Why these environments were selected, and if the underlying environments have enough customisability to generate curricula that are useful

---

> ### Author Response · Authors · 2023-11-21
>
> We thank the reviewer for insightful and helpful comments! Here’s our response to questions and weaknesses raised in the reviewing process.
>
> 1. Implementation of Blind Agent and Design Choices
>
> The action sequences in a blind agent are first initialized using a uniform distribution. Following is the code used to initialize blind agents at the start of the training;
>
>     # max_ep_len : Maximum length per episode of the given environment
>     # action_dim : dimension of the action space.
>     import numpy as np
>     def initialize_blind_agent(max_ep_len,action_dim):
>         timestamp_count = np.random.random.randint(1,max_ep_len)
>
>         genome = []
>         for _ in range(timestamp_count):
>             actionCallout = []
>             actionCallout.append(np.random.randint(max_ep_len)) #Time
>             actionCallout.append(np.random.uniform(low=-1,high=1,size=action_dim)) #Action
>             genome.append(actionCallout)
>
>         return genome
>
> During training, we record the outcome of each blind agent action sequence in the population. Using mutation and crossover, we use this data to genetically optimize the generate the population of blind agents based on the fitness function p(ξ) as explained in Section 3.3. To the best of our knowledge, we are the first paper to train a competitive multiagent RL policy with a genetically optimized opponent.
>
> The key feature of a blind agent is that it is a genetically optimized opponent policy that is blind to adversarial noise. For example, if we apply noise to the opponent RL policy, the underlying RL policy network is still vulnerable to noise and we get results similar to the NoBlind in the ablation studies as shown in Section 5.4. If we use random policy instead of genetically optimized policy, it works in simple environments such as Pong. However, in a complex environment like the ACM, a random policy would easily crash to the ground. The training suffered due to the long interim phase where the RL policy is winning random policies by staying in the air longer than the random policy that keeps crashing to the ground but has not yet learned useful tricks that facing another RL agent would be interesting. Using genetically optimized blind agents smooths the transition between training against open-loop opponents to avoid overfitting and training against closed-loop RL agents to know what to do against challenging policies.
>
> 2. What if we run the algorithm for a much more customizable environment?
>
> We chose Pong, Volley, and ACM to show how our algorithm performs in various different environments ranging from simple 1D movement space with simple game-style physics to complex 3D movement space with complex aerodynamic models. We agree that seeing how our algorithm works in a much more customized environment. We include the results from the the BipedalWalkerHardcore benchmark (Brockman et al, 2016). BipedalWalkerHardcore is an environment where an agent must learn to traverse through various kinds of obstacle courses. Every terrain and obstacle feature in the environment is customizable and it takes up to 300 integers and floats to fully describe an obstacle course, making the scenario space quite large. The current state of the art in this benchmark is the Genetic Curriculum (Song et al), which has a failure rate of 3.96±0.37%. The performance of our proposed algorithm is 4.27±0.99%, within the standard error from the state of the art.
>
> Again, we sincerely thank the authors for their insightful and helpful comments and please let us know if you have any questions or comments!

---

> ### Comment · Reviewer_wvYt · 2023-11-22
>
> I thank the authors for taking the time to answer my questions.
>
> Point 1. Thank you for clarifying this. However, I still think there needs to be work done within the paper itself in terms of clarifying and motivating the purpose of the blind agent. This also seems to be picked up by the other reviewers.
>
> Point 2. Thank you for adding the additional result. However, it is currently not enough for me to increase my score. For example, I need to know further implementation details of the experiment (I am mostly unfamiliar with the exact task), how many seeds was this run over (a 0.99 standard error seems large), why was failure rate the used metric (a quick search for this environment seems to mostly report mean reward values) etc...

---

> > ### Author Response · Authors · 2023-11-22
> >
> > We thank the reviewer for the helpful feedback! Here's our response.
> >
> > Point1. Thank you! We'll include this point in our final draft.
> >
> > Point2. Sure! We can explain more about the BipedalWalkerHardcore problem, and all the stated explanations will be included in our final draft.
> >
> > BipedalWalkerHardcore (https://www.gymlibrary.dev/environments/box2d/bipedal_walker/) is an Open AI Gym environment powered by a 2D physics simulator, Box2D. The ego agent is a bipedal walker who has to traverse through various obstacles consisting of stairs, stumps, and pitfalls. Raw encoding of the environment can require more than 300 integers and floats to describe an obstacle course. The agent has a 24-D observation space consisting of the position, velocity, and angle of each component of the robot, as well as a LIDAR scan in front of the robot. It has 4D continuous action space, which controls how much torque is applied to each of the robot's joints. We chose this environment as this is the environment used for the original Genetic Curriculum paper, one of the benchmarks for our paper.
> >
> > For our experiment, we used the same set of hyperparameters used for the Genetic Curriculum paper. We trained 5 seeds on the task, which is as same as the number used for the Genetic Curriculum paper. Reward-wise, Genetic Curriculum reports 304.33±1.65 whereas we performed better in this regard and got 311.21±3.56.
> >
> > We thank the reviewer for taking the time to read our response! If you feel like there was a point not well addressed in our rebuttal, please let us know.

---

### Official Review · Reviewer_NHU7 · 2023-11-02

**Soundness:** 3 good
**Presentation:** 3 good
**Contribution:** 2 fair
**Rating:** 5
**Confidence:** 3

**Summary:**

To learn policies in complex and unstable MARL environment and game-theoretic setup, this paper presented a curriculum learning method using the Genetic algorithm. Its main contributions include 1) population-wide genetic operations (crossover) 2) introducing a regret to accommodate the difficulty level of genetically generated scenario, and 3) continuously optimized open-loop opponents to stabilize early learning. Ablation and empirical study demonstrated the effectiveness of the proposed algorithm and design choices.

**Strengths:**

1.	The paper studies an important problem in MARL. It is well motivated and organized

2.	Empirical evaluation on three domains demonstrated the effectiveness of the proposed method comparing to a few baselines

**Weaknesses:**

1.	Some related work are missing (see question 1 and 2)

[1] Evolutionary Population Curriculum for Scaling Multi-Agent Reinforcement Learning
Qian Long, Zihan Zhou, Abhibav Gupta, Fei Fang, Yi Wu, Xiaolong Wang, ICLR2020

[2] Learning Multi-Objective Curricula for Robotic Policy Learning
J Kang, M Liu, A Gupta, C Pal, X Liu, J Fu, CoRL2022

2.	The study is mostly empirical, no theoretical analysis is provided regarding convergence, computational and sample complexity
3.	There are some claims without clear justification (see question3)

**Questions:**

1.	To me, the purpose of this work is to use curriculum learning to address the complexity of policy learning for multiagent tasks. There are two major dimensions for characterizing the complexity of a problem, one is from the large state space and its associated dynamics(exogenous), the other one is due to the interaction among multiple agent. It seems to me that this paper is mainly focus on the first aspect of the complexity, which is not unique to the multiple agent problem. Can the same method be applied to single cases?
2.	How about the second source of complexity. In other words, how the method will perform in there the number of agent is large? Ref [1] already provided a solution method by using evolution algorithm which is simpler than genetic algorithm, except that there is no crossover is considered.  The value of this work would be significant enhanced if such as aspect is considered.
3.	In section 3.4 it is mentioned that “instead of searching for the global optimum, the student agent will often exploit…”. It is unclear to me what does “global optimum” mean? Do you mean pareto optimum?
4.	What are u and \delta in Algorithm1?
5.	In section 4.3, first paragraph, “…intensive algorithms like PONG…” What is PONG?
6.	Given the description in the second paragraph of section 3.1, do you assume that the agents are homogenous?
7.	There are quite a few typos needs to the fixed, please proof read.

---

> ### Author Response · Authors · 2023-11-21
>
> We sincerely thank the reviewer for giving helpful comments on how to improve our paper! Here’s our more detailed response to weakness and questions made in the review.
>
> Weakness 1 : We thank the reviewer for mentioning the paper! We agree that these papers highlight important ideas in curriculum learning relevant to our research and will certainly include the citations in our publication.
>
> Question 1 : To verify how well our algorithm performs to complexity in large state space, we ran our algorithm on the BipedalWalkerHardcore benchmark (Brockman et al, 2016) as used in the Genetic Curriculum (Song et al, 2022). In this environment. BipedalWalkerHardcore is an environment where an agent must learn to traverse through various kinds of obstacle course. We felt that this is an interesting case study for a genetic algorithm-based curriculum generator as obstacle courses are represented as scenario encoding of various length ranging from 20 to 300D. Using same set of hyperparameters, scenario encoding, crossover and mutation function, our proposed GEMS achieved failure rate of 4.27±0.99%. whereas Genetic Curriculum, the current state of the art algorithm in the benchmark, got 3.96±0.37%
>
> Question 2 : We too agree that it would be interesting to see how scalable our algorithm is in terms of number of agents. We are currently running experiments based on multiplayer environments similar to ones used in [1] where there are 4, 16, and 32 players in total. In these environments, agents will be playing a cat and mouse game where the one side is trying to chase the other side whereas the other side is trying to evade and avoid being chased. While we’ll have to wait until tomorrow to complete the training, the current training progress looks promising. We’ll post the results when they are ready before the end of the rebuttal period.
>
> Question 3. In the section 3.4, we were referring to the phenomenon where the agent being trained in competitive settings tends to overfit to the adversary it saw during the training and fails to generalize to a wider population of opponents. This trend with deep neural networks has been reported in various supervised and reinforcement learning papers, such as;
> - Understanding Robust Overfitting of Adversarial Training and Beyond, Yu et al, 2022
> - Efficient Adversarial Training without Attacking: Worst-Case-Aware Robust Reinforcement Learning, Liang et al, 2022
> - Overfitting in Adversarially Robust Deep Learning, Rice et al, 2020
> - Robust Reinforcement Learning via Genetic Curriculum, Song et al, 2022
>
> We will include our response to your Question 3 in the final draft.
>
> Question 4. μ an integer value recording whether the policy π has won, lost, or tied in the given scenario ξ. δ is the value of the regret of policy π in the given scenario ξ. We’ll fix these typos in the final draft.
>
> Question 5. PONG refers to the Pong environment. We’ll fix this issue for the final draft.
>
> Question 6. In our paper, we only considered the cases in which the role of the ego agent is same as the role of the opponent agent. The reward function, observation space, action space, and dynamics are the same for both ego and opponent agent.
>
> Question 7. We acknowledge the need and will clean up the mistakes for the final draft.
> We again thank the reviewer for helpful feedback! We’ll include these details and clarifications for our final draft. Please feel free to let us know if you have any questions!

---

> ### Author Response · Authors · 2023-11-22
> **Scaling with respect to the number of players**
>
> Some of our reviewers wondered how scalable our algorithm scales with respect to the number of players in a game. We therefore ran the SimpleTag environment in the PettingZoo (Terry et al, 2021) benchmark.
>
> In this predator-prey environment. There are n-players taking the two teams. In the ‘Good’ team, players get a negative reward (-10) every time they run into members of the ‘Bad’ team. The members of the ‘Bad’ team are slightly slower than the ‘Good’ team and get a positive reward (+10) every time they run into the good team. To make the environment more interesting, there are two large obstacles occupying the map, forcing the agents to maneuver around the obstacle. Each agent observes its own speed, location, the location of the obstacle, the location of the other agents, and their velocities. To help with learning, there’s a sparse dense reward function associated with chasing and evading.
>
> In the ST4 benchmark, there are a total of 4 players in the game, 2 playing as ‘Good’ and 2 playing as ‘Bad’. Trained for 4e5 steps and with 5 seeds, the performance of trained algorithms against each other is as follows;
>
>
> ALGORITHM      |FSP                       |PSRO                    |GC+FSP                |SPDL+FSP           |MAESTRO            |GS4
>
> MEAN REWARD|-14.516±0.517       |-20.025±0.746       |-26.054±1.232       |-27.13±0.586        |-22.722±0.765       |-12.624±0.673
>
>
> In the ST8 benchmark, there are a total of 16 players in the game, 8 playing as ‘Good’ and 8 playing as ‘Bad’. Trained for 2e6 steps with 5 seeds, the performance of trained algorithms against each other is as follows,
>
>
> ALGORITHM      |FSP                      |PSRO                   |GC+FSP               |SPDL+FSP          |MAESTRO           |GEMS
>
> MEAN REWARD|-1.353±1.681        |-6.472±1.533        |-0.002±1.286        |0.942±2.202         |4.193±1.515         |6.403±1.52
>
>
> In all experiments regarding scalability towards multiple players, our agent has outperformed all baselines. We'll add 5 additional seeds to report the results with 10 seeds for our final draft.
>
>
> In our experiments, we did not implement the algorithm from [1] Evolutionary Population Curriculum for Scaling Multi-Agent Reinforcement Learning Qian Long, Zihan Zhou, Abhibav Gupta, Fei Fang, Yi Wu, Xiaolong Wang, ICLR2020. While we find the paper interesting and we will certainly include the papers in the related works on how the genetic algorithm has been used for multiagent training, we found that for the following reasons, it is better to focus on the baselines we already included.
>
> First, the success of [1] is attributed to various reasons not related to the aspect of curriculum learning we are interested in exploring. For example, [1] uses a population-invariant Q function which allows the agent to scale quickly to different numbers of players. Also, while [1] and our proposed algorithm use curricular ideas, [1] is different in the sense that it uses a population of ego agents trained simultaneously and uses crossover and mutation to choose which of the agents should be carried over to the next round. On the other hand, ours, along with other baselines included in our paper such as PSRO and MAESTRO, only trains one agent, keeps a population of opponent policy as saved fixed checkpoints of the ego agent, and focuses the curriculum selection as a selection of opponent policy and (in the case of ours, MAESTRO, SPDL+FSP, GC+FSP) the selection of environmental parameters. For the scope of this research, I think it would be better to explore on curriculum aspect of choosing the right opponent and environmental parameters.
>
> Second, implementing [1] is quite challenging. While there are publicly available codes for [1], they are quite old and some of the packages are outdated. Furthermore, the publicly available codes are not well documented and several comments are missing, making it difficult to implement and integrate with the rest of our codebase within the timeframe.
>
> Thank you so much for your helpful feedback! Let us know if you have any questions or would like to point out what our rebuttal missed. If you think our rebuttal looks acceptable, please feel free to adjust our score accordingly.

---

> > ### Comment · Reviewer_NHU7 · 2023-12-04
> > **response to rebuttal**
> >
> > Thanks for the authors' responses to my comments and questions, which really help clarifying some of my confusions and concerns. I also appreciate the authors' effort in putting additional results to address my concern over the scalability over the number of agents. Hence, I would like to increase my score by one point. However, I still think the paper is slightly below acceptance threshold given 1) its current form (missing compressive literature survey as pointed by other reviews as well); 2) this study is mostly empirical, no theoretical analysis is provided regarding convergence, computational and sample complexity; 3) only one performance metric (mean reward) is reported above (for competitive problem, how can only mean reward be used? can the method guarantee any pareto optimality? or you are just interested in ego agent?).

---

### Meta-Review · Area_Chair_teWB · 2023-12-05

**Metareview:**

All reviewers agreed that this paper was not quite ready to be accepted in its current form. In particular, the method seemed too ad hoc, and unlikely to generalize to other situations. There were also concerns about limited novelty.

**Justification For Why Not Higher Score:**

None of the reviewers were terribly enthusiastic. All gave. score of 5.

**Justification For Why Not Lower Score:**

N/A

---

### Decision · Program_Chairs · 2024-01-16

Reject